# Small-Scale Analysis of Characteristics of the Wildland–Urban Interface Area of Thessaloniki, Northern Greece

Petros Ganatsas *, Nikolaos Oikonomakis and Marianthi Tsakaldimi

Laboratory of Silviculture, School of Forestry and Natural Environment, Aristotle University of Thessaloniki, P.O. Box 262, 54 124 Thessaloniki, Greece

* Correspondence: pgana@for.auth.gr

**Abstract:** In the past few years, the continuous expansion of urban development has created mixed forested, agricultural, and urban areas. These areas are called the wildland–urban interface (WUI), and they are characterized by increased human activities and land-use conversion, and they usually contribute to a high risk of wildfire occurrence. In the case of the peri-urban areas of Thessaloniki city, an effort was made to map, classify, and describe this wildland–urban interface, using Sentinel-2 satellite images of the area and very large scale orthophotos (VLSO) for the human settlements. Object-based image analysis (OBIA) was applied to classify landcover, combined with analysis of field data. The results showed that the WUI area in the city of Thessaloniki appears to the north and east of the city and covers an area of 2203.98 ha. The main characteristics affected by the ecological conditions of this area are the building (or human infrastructure) density, type, and the structure of forest vegetation. Human population pressure was found to be greatly differentiated between WUI areas belonging to different municipalities, the most affected was the municipality of Thessaloniki. A set of fire prevention silvicultural treatments are suggested for mitigating the fire danger in the area, accompanied by appropriate human awareness actions and the involvement of the local society. These measures include the reduction in crown bulk density and increase in crown base height through pruning (at least to 1/3 of total tree height), and low thinning, aiming to ensure that tree crowns of mature *Pinus brutia* trees are not in contact with one another. Both in the young *P. brutia* forest and the evergreen vegetation areas, thinning, pruning, and vegetation clearing is recommended adjusted according to each ecosystem.

**Keywords:** wildland–urban interface; OBIA; GIS; forest fires; peri-urban forest; urban ecology





## 1. Introduction

In recent years there have been strong trends in the expansion of residential development (controlled or not), to the detriment of forest or rural areas. One result of these trends is the creation of areas where forest vegetation coexists with buildings or various human infrastructures. The term suburban or peri-urban zone cannot be used to precisely describe these areas, which in the international literature are specified as forest-settlement mix zones; they are called wildland–urban Interfaces (WUIs) [1]. These areas have been given much attention during the last years, mainly due to the great environmental problems that arise within these areas, one of which is the great danger from wildfires. Especially for the countries of the Mediterranean basin, the continuous expansion of the wildland–urban interface areas results in an increasing vulnerability to forest fires [2]. However, we must note that the term wildland–urban interface has been differentially interpreted when used under different perspectives, and that its main use is now in the context of wildland fires [3]. In the United States, where the importance and need for proper management of these areas has long been recognized, the following definition is commonly used: an area of forest-settlement mix (wildland–urban interface) is characterized as the one where the buildings are in contact or mixed with forest vegetation [4]. The specific term moves in the same spirit at the European level [5].

Many efforts have been made to determine, describe, and map WUI areas in different parts of the world, mainly in the USA and Europe, at different scales, producing local, regional, or national maps, based on the scope of the study [1]. Maps at a local scale include more details and usually provide information on methods for the coexistence of wild vegetation and urban land uses [6–8].

However, WUI areas are among the areas most vulnerable to wildfires in many countries around the world (e.g., the extreme wildfire in Mati, Attica, Greece in 2018, where 102 citizen fatalities occurred). The identification, mapping, and determination of vulnerable areas and ecosystems to fire are of high priority in wildfire management policies in many countries. Thus, the development of special fire management plans that can be applied in a way that will effectively prevent large and catastrophic fires are of urgent need [9]. These plans are developed based upon the information available concerning the characteristics and the spatial distribution of these important zones of WUIs. In recent years, there have been a wide range of methods and applications, including geographic information systems and remote sensing data and methods. Although these are stand-alone methods, they are accompanied by field data that are used in WUI determination and mapping. Additionally, several modeling tools are used for the simulation of fire behavior, which could help in effective planning and the management of any possible fire event. For example, Molina et al. [10] developed an ignition index for application at the wildland–urban interface (WUI) via the integration of several components of fire risk (fuel moisture content, climatic data, physiographic parameters, and vegetation flammability). Mitsopoulos et al. [11] developed a spatially explicit estimation of wildfire potential in the WUI of Attica in Greece by assessing the wildfire risk using fuel data, fire alternative simulations, and different scenarios in burning conditions. However, basic spatial environmental and ecological characteristics are in all cases greatly needed, since ecological characteristics, such as vegetation type, dominant tree species, and vegetation height play an important role in fire behavior [12]. In practice, identifying, mapping, and describing WUI sites require specific data on human presence, vegetation, and spatial expansion zone. The human presence is usually determined based on the density of settlements or population, the vegetation cover, while some difficulties occur to the extent that this zone extends on both sides to purely forest or urban areas [3].

Though clear methodological approaches for mapping WUI exist [13], the methods do not provide the ecological precision required for effective management interventions. Furthermore, an estimation of urban pressures could help in strategic management options. In the frame of the present study, an analytical attempt was made to record, classify, and map the suburban zone of Thessaloniki that has the characteristics of WUI areas, with a basic-tool analysis of satellite images, their processing with geographic information systems (GIS), and field data analysis. Considering the high human pressure and the associated fire risk, we emphasized examining the specific structural characteristics (ecological and human) of the studied area.

The specific aim of this study was the development of a combined approach for the determination and evaluation of the wildland–urban interface (WUI) area of Metropolitan Thessaloniki, the second largest city of Greece, as a novel analytical approach that can result in a more effective fire management in the area. We hypothesized that an analytical mapping of human infrastructures, ecological characteristic variation within a WUI, and an estimation of population pressure on the different part of the WUI area, could help towards this direction. The ultimate goal was the utilization and design of specific fire-protection measures, determined spatially in prioritized land areas.

## 2. Materials and Methods

### 2.1. Description of Study Area

The climate of the studied suburban zone of Thessaloniki is characterized as semi-arid, Mediterranean, with cold winters and high temperatures during the summer. The area receives 420 mm annual precipitation, while the dry season lasts for four months, from

mid-May to mid-September. The climatic conditions (especially the prolonged dry period) together with the types of the vegetation (Mediterranean-type ecosystems) of the area entail a high risk for wildfires [14]. Half of Thessaloniki's peri-urban forest was burnt in 1997, causing great ecological damage to the area and the city. Furthermore, a small number of forest fires occur in the area during the summer period every year, which are suppressed in their initial stages.

### 2.2. Determination and Mapping of WUI Areas

The study was based on updated cartographic data from the free public site of the official Greek Cadastre [15], and the frequent, precise, and accurate Sentinel-2 mission images which are distributed free of charge. Buildings were systematically recorded (and digitized as points) by photo-interpreting methods using the most recent backgrounds of Google Earth and the Greek Cadastre.

The WUI areas are usually mapped and classified according to the configuration of building density and vegetation structure (trees, shrubs, or grass) based on recent satellite images [16]. Thus, a recent Sentinel-2 image background was employed to represent the wider area of Thessaloniki and analyzed in an object-oriented image analysis (OBIA) environment, using the software eCognition [17]. The images from which the background was created were selected based on the lowest possible cloud cover (images acquired on 8 May 2021). The Sentinel-2 mission consists of two polar-orbiting satellites launched by the European Space Agency [18], carrying a multispectral instrument (MSI) sensor and delivering data at spatial resolutions between 10 and 60 m, covering a wide part of the electromagnetic spectrum, from visual to infrared, and a temporal resolution of 5 days.

The images were processed at Level 2A, so they already provided geometrically and atmospherically corrected at bottom of atmosphere (BoA) reflectance. For the image analysis, only the spectral bands at 10 and 20 m spatial resolution were employed. Here, OBIA was selected because of the several advantages compared to traditional pixel-based approaches when applied to high spatial resolution data [19]. This method was effective with the within-class variability, and since the classification was carried out on the object unit (and not on the pixel unit), various shape, texture, and context characteristics can be employed in the classification process [20]. Furthermore, OBIA was used to avoid the pixelized (salt and pepper) representation of land cover classes which is often observed in pixel-based approaches, while the final classification product was integrated directly into a vector-GIS for further analysis [21]. The foremost part of OBIA was a segmentation of an image into multi-pixel image object primitives. The result of segmentation was controlled by user-defined scale parameters, selected according to the size and characteristics of the expected objects [22] that were adapted accordingly to the characteristics of the OBIA objects of the current study.

Within the suburban area of Thessaloniki, the forest area was determined by applying an object-based classification assisted by thematic layers of data, such as (1) delineation of forest and grassland maps by the Hellenic Cadastral Organization, (2) orthophotomaps (forest vegetation maps from the Hellenic Forest Service), and (3) the dominant leaf type (DLT) product of the high resolution layer (HRL) products of the reference year 2018 from the Copernicus Land Monitoring Services (CLMS) [23]. These data were used to assist in the classification and to avoid misclassifications. Corine 2018 for Greece also used to verify and assess the classification. Rules were defined to determine the main classes of interest using spectral characteristics and object shape characteristics. Apart from the original spectral bands, the spectral indices were also calculated and used in the analysis, such as normalized difference vegetation index (NDVI), enhanced vegetation index (EVI) [24], and a band ratio for built-up areas (BRBA) [25].

$$\text{NDVI} = \frac{\text{BAND 8} - \text{BAND 4}}{\text{BAND 8} + \text{BAND 4}} \tag{1}$$

$$\mathrm{EVI} = 2.5 \left( \frac{\mathrm{BAND\ 8} - \mathrm{BAND\ 4}}{\mathrm{BAND\ 8} + 6 * \mathrm{BAND\ 4} - 7.5 * \mathrm{BAND\ 2} + 1} \right) \tag{2}$$

$$\mathrm{BRBA} = \frac{\mathrm{BAND\ 3}}{\mathrm{BAND\ 8}} \tag{3}$$

Afterwards, a transition zone 500 m wide [26,27] outside the city borders was mapped regardless of the existence of any building within this zone (Figure 1). This width value was based on the distance that a fire spot can run in the fire front in the case of wildfires (spotting) [28]. A similar approach for distance determination is reported by Summerfelt [29]. This was set as the first step for the determination of the WUI area of Thessaloniki. The WUI area was distinguished in two categories, as follows: (1) WUI category I was considered the area within an external buffer zone of 500 m outside the city borders; (2) WUI category II was considered as the areas outside the WUI category I zone which had a housing density over 1 house/building per 16 ha. A digital elevation #,odel (DEM) was also used [30] and processed properly with GIS functions concerning the production of surfaces of altitude, slope, and aspect, to be able to assess those factors of the WUI area. Additionally, for the determination WUI category II, a building density surface was produced to map density information and distinguish the WUI category II areas (Figure 1).

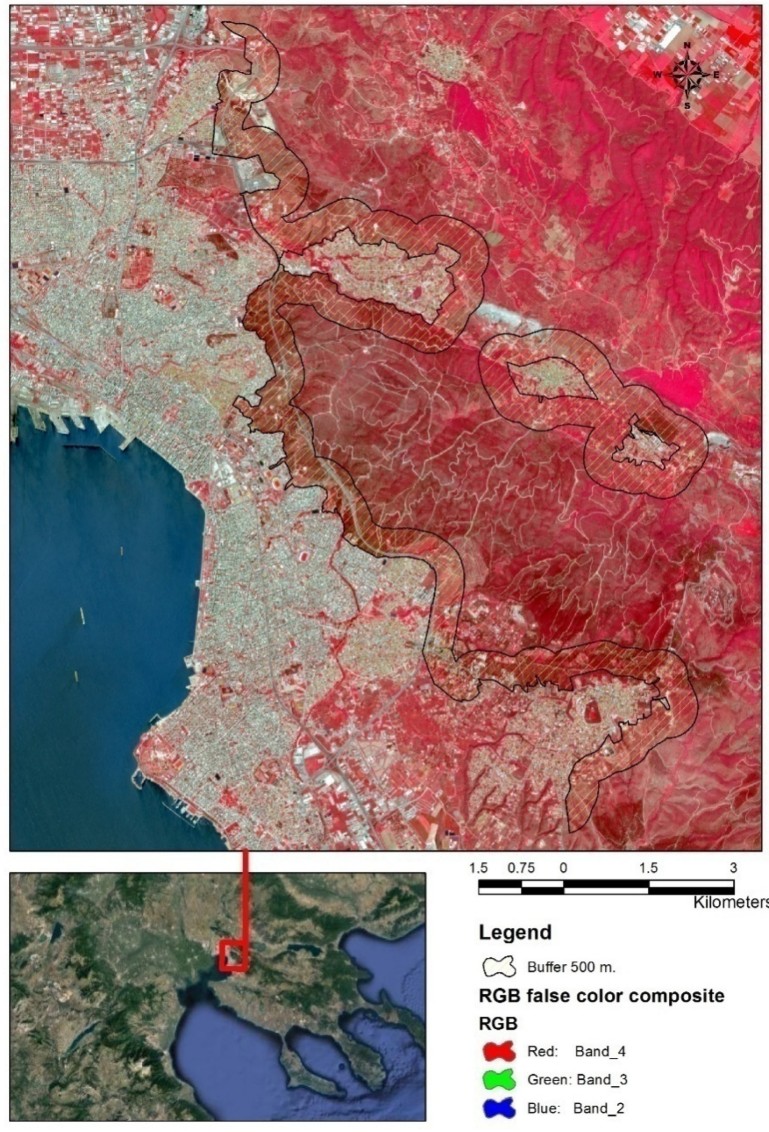

**Figure 1.** Sentinel-2 images used for the land cover mapping of the study area (May 2021). An external buffer (500 m) outside the city borders is shown, determined as **WUI category I**.

Then, the whole suburban area was categorized according to the type of existing vegetation by using an assisted OBIA classification with the available thematic layers and the spectral signatures of the forest areas in the infrared and near-infrared spectrum. Additionally, field measurements confirmed the results of vegetation classification. Consequently, a land-cover map of the wide area of Thessaloniki was produced (Figure 2). All classifications were verified with the VLSO dataset and Google Maps using a set of 100 random samples per class with the help of the "sampling design tool", an add-on in ArcGIS [31].

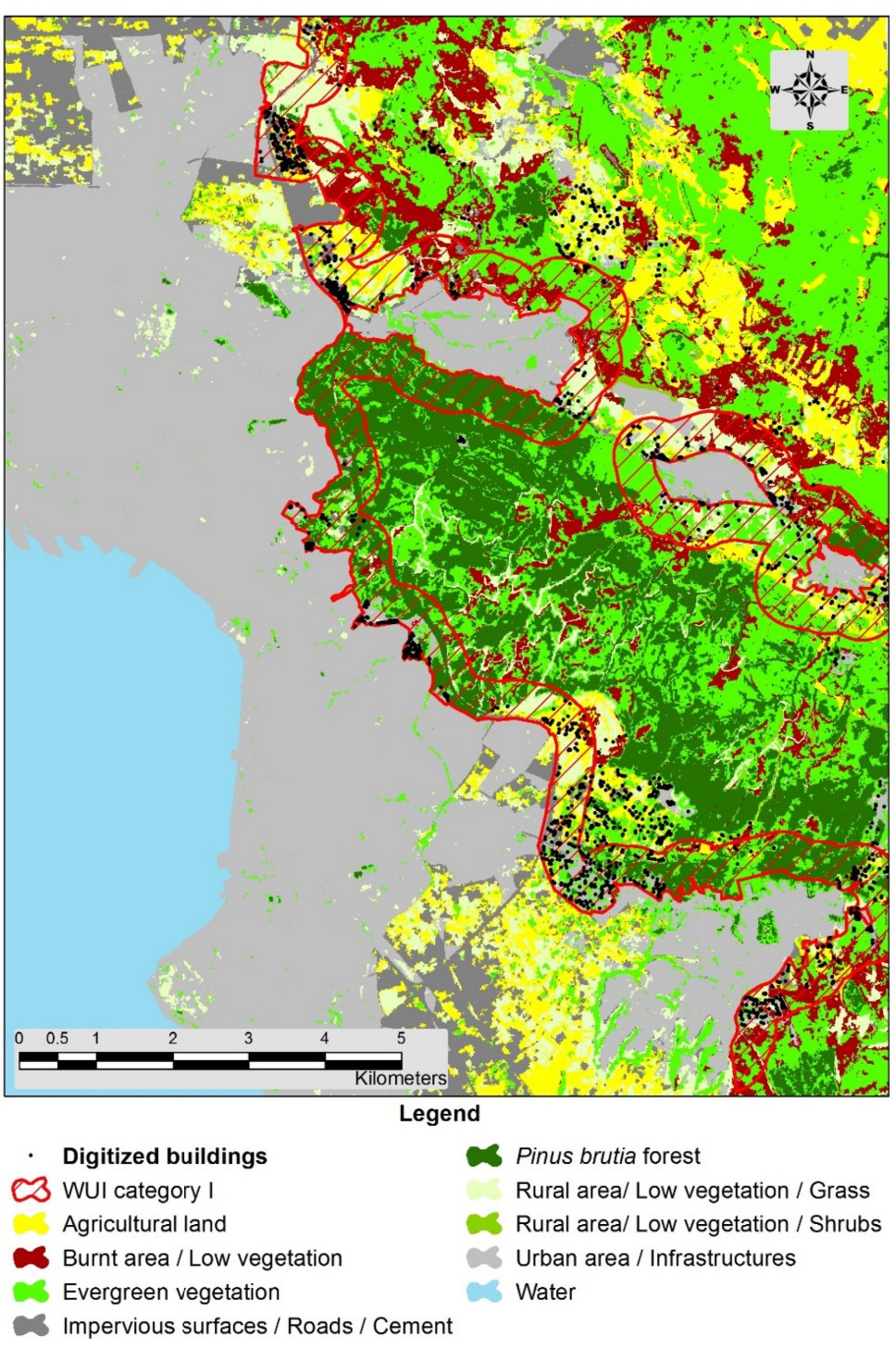

**Figure 2.** OBIA classification results of study area using Sentinel-2 image background (pixel size 10 m). **WUI category I** (buffer 500 m outside urban areas) is also presented. Dots represent buildings.

Buildings were defined using the VLSO RGB background provided by the Hellenic Cadastral Organization (pixel size 20 × 20 cm in the city and 50 × 50 cm in other areas) of the year 2015 and verified and supplemented with the latest Google Earth background of the year 2019. All building structures with tiled roofs and terraces were recorded as buildings, avoiding stables and other non-permanent constructions. The result of the systematic recording of buildings was a file consisting of 1920 points (1435 of them were within the buffer zone of 500 m around the city borders), which represent the geographical position of each building. Afterwards, the building density surface was created for the whole suburban area, using the point density and the kernel density functions in the GIS environment using a search radius of 56.42 m, which represents an area of 1 ha for each point. Kernel density function was used to produce an interpolated surface to be able to classify also with decimal values and to calculate areas of specific density.

Areas outside WUI category I (at the forest side) with housing density over 1 house/building per 16 ha was mapped as WUI category II. Afterwards, a subdivision was applied in both WUI categories according to the density values [32], which categorized WUI areas using two criteria, namely (a) house/building density, and (b) vegetation type. In this study, according to the first criterion (building density), four classes were distinguished, as follows: (1) density values from 0 to 0.0625 (1 building/16 ha), (2) density 0.0626–2.5 buildings/ha, (3) density 2.5001–6.0000 buildings/ha, and (4) density > 6.0000 buildings/ha (Figure 3). However, the first class was recorded only in WUI category I. This classification follows the general approach of Lampin-Maillet et al. [32], who classified the dwellings into three types—isolated dwellings, scattered dwellings, and clustered dwellings—using spatial criteria, and based on the distance between buildings, the size of clusters of buildings, and housing density. In our study, we simplified the method. The classification was made only upon the buildings' density.

### 2.3. Estimation of Human Population Pressure in the WUI Area of the City of Thessaloniki

The digital cartographic backgrounds, the borders, and the population evolution of all the adjacent municipalities, were extracted from the archives of the Hellenic Statistical Authority. Then, based on the above data, we analyzed the human population pressure [33] in the WUI area of the city of Thessaloniki, using the findings of a simple questionnaire that was electronically distributed via the Internet to the citizens of the area of Thessaloniki. The questionnaire prepared aimed to estimate the number of people visiting (human population pressure) the peri-urban forest and the WUI area [34]. A total number of 400 questionnaires were collected (100 per municipality) and the data were statistically analyzed for determining the number of human visits per year per hectare in each municipality area. The main question-set concerned the frequency in visiting the peri-urban forest per week/month/year, and the distribution of these visits within the year. The classification was as follows: (a) every day, (b) two or three times per week, (c) two or three times per month, (d) once per month, (e) two or three times per year, (f) less than once per year, (g) never [35].

### 2.4. Estimation of Ecological Characteristics in the WUI Area of the City of Thessaloniki

Finally, for the estimation of the ecological characteristics of the WUI area of Thessaloniki, in each of the main forested land types, ten sample plots were taken in each vegetation class, as follows: in the pine forest (*Pinus brutia* forest), in the shrubland (broadleaves evergreen vegetation), and in the burned (in 1997) forest area. The size of the plots was 500 m$^2$ in the pine forest, and 100 m$^2$ in the other two vegetation categories. In each plot all the woody species were recorded for their DBH (diameter at breast height, in cm), and total height (accuracy 0.1 m). The canopy cover was visually estimated separately by two experts with long experience, in order to avoid subjectivity, for overstorey, shrubstorey, and understorey, and the stand structure was estimated based on stem distribution. A soil profile and soil samples from two depth layers (0–15 cm and 15–30 cm) were taken from each plot (center point) and were transported to the laboratory for analysis. Soil depth was

estimated in the field, while laboratory analysis carried out at the Laboratory of Silviculture AUTH included organic matter content, pH determination, total nitrogen, and mechanical analysis (particle size distribution).

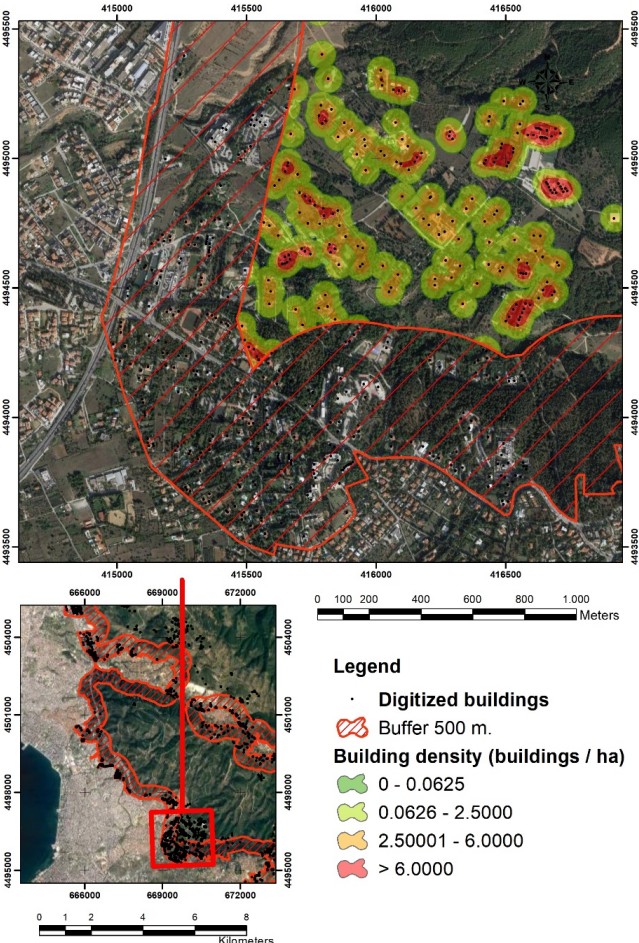

**Figure 3.** Building density surface has been produced for WUI category II. Classes with a density over 0.0625 (or 1 building/16 ha) and outside the buffer zone of 500 m represent WUI category II.

### 2.5. Statistical Data Analysis

Statistical analysis of the collected ecological and human population data was performed using the SPSS® software v. 23.0 (SPSS Inc., Chicago, IL, USA). One-way ANOVA was used to test any significant differences between the plant (canopy cover per woody species, and totally, %) and soil characteristics (depth, soil organic matter, pH, and nitrogen content for each soil stratum) in each vegetation type observed in the study area, as well to analyze data on the human population pressure, in terms of total number of visits per year and hectare. The Waller–Duncan test was conducted to compare the computed mean values. Statistical analyses were conducted at the α = 0.05 significance level.

### 3. Results

### 3.1. WUI Area of Thessaloniki

The results of the validation of classification showed a satisfactory accuracy. This was because of thematic layers that helped the delineations. The general classification from Sentinel-2 image background was precise, because it was easy to differentiate objects with "*Pinus brutia* forest" from "Urban Area and Infrastructures" due to the high mean values of the vegetation indices (NDVI and EVI) in "*Pinus brutia* forest" and low values in "Urban Area and Infrastructures", respectively. The overall accuracy value was 83%, as the confusion matrix shows (Table S1).

The suburban area of Thessaloniki that can be characterized as being a wildland–urban interface (**WUI category I**) according to the criteria adopted occupies an area of 2203.98 hectares (Figure 4). We also found only a very small area (225.57 ha) belonging to **WUI category II**. The WUI area of Thessaloniki is in the northeastern border of the city. The WUI's elevation varies between 56–494 m asl, and it is located 1–8 km from the sea. It appears in all slope aspects, with the largest part existing in the southwest (26.47% of the WUI area) or west (20.42% of the WUI aspects), and mainly in moderate slopes (0–35%). Furthermore, low slope values were present (0–15%) in 88.68% of the total WUI area, while slopes >30% existed in only 0.16% of the WUI area.

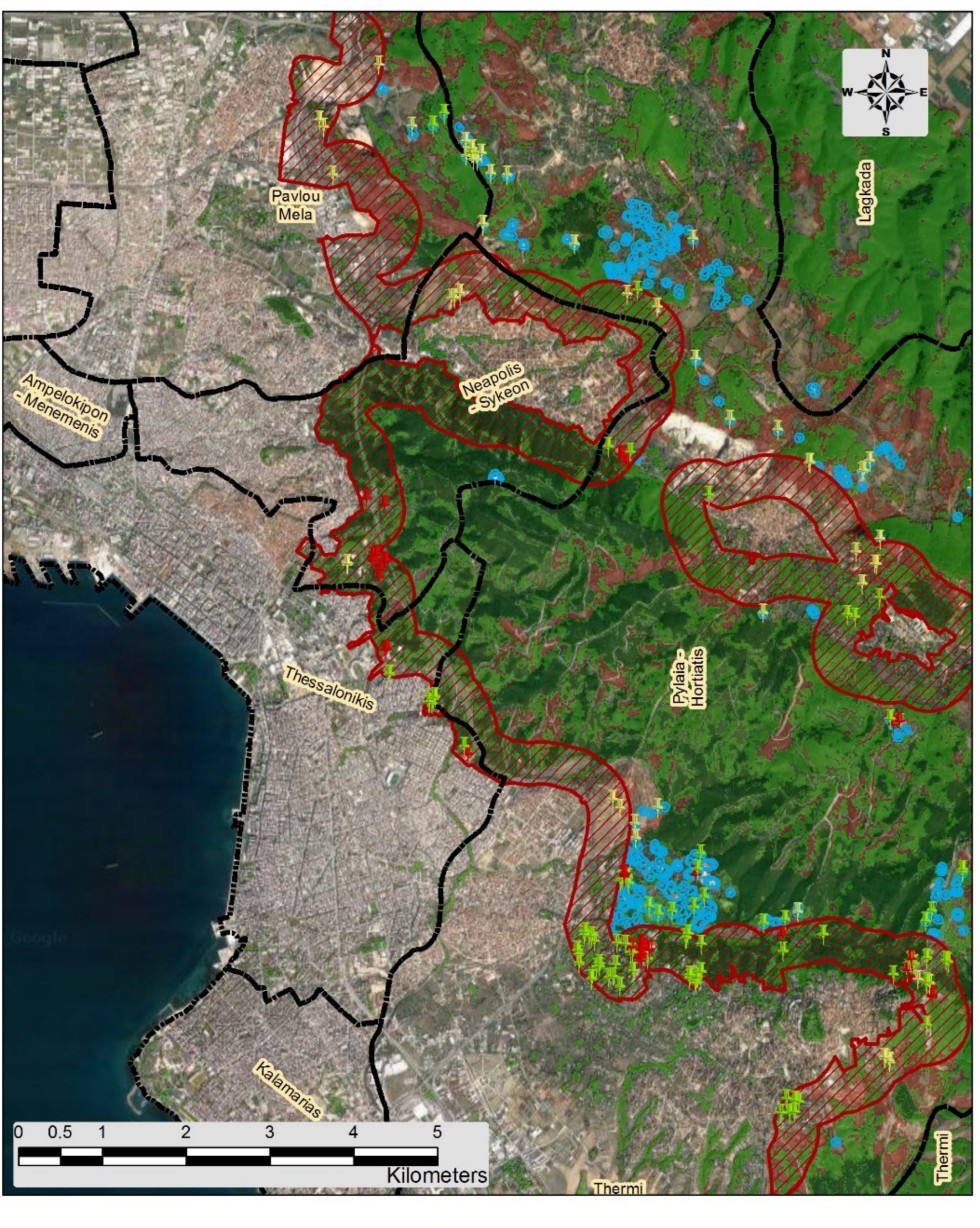

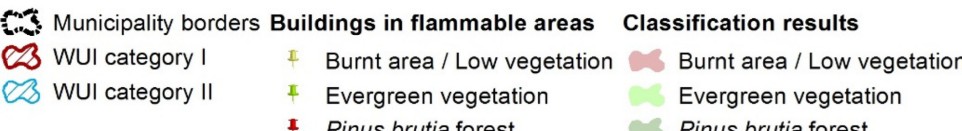

**Figure 4.** Map showing the two WUI categories, the vegetation classes within the WUI areas, and the buildings in areas over 2.5 buildings/ha.

Concerning the type of vegetation land uses within the WUI area, the analysis of land cover types showed that, within the wider area of Thessaloniki, nine vegetation classes of land cover types were identified (Table 1). The largest percentage was occupied by the areas with forest and other natural low vegetation (*Pinus brutia* forest, evergreen vegetation, burnt area/low vegetation, rural area/low vegetation/grass, and shrubs) which occupy a total of 78.16% of the total WUI area, while the areas with human activities (agricultural land, impervious surfaces/roads/cement and urban areas/infrastructures) occupy a total of 21.84% of the total WUI area.

**Table 1.** Land cover types in the peri-urban area of Thessaloniki (in the 500 m buffer outside city borders) WUI category I.

| Classes | Value/Code | Land Coverage (%) |
|---|---|---|
| Agricultural land | 1 | 8.8 |
| Burnt area/low vegetation | 2 | 13.22 |
| Evergreen vegetation | 3 | 27.79 |
| Impervious surfaces/roads/cement | 4 | 5.51 |
| *Pinus brutia* forest | 5 | 22.46 |
| Rural area/low vegetation/grass | 6 | 14.33 |
| Rural area/low vegetation/shrubs | 7 | 0.36 |
| Urban areas/infrastructures | 8 | 7.55 |

The analysis of the density of buildings showed that, within the WUI zone (WUI category I), the building density ranges between 0 and 50 buildings per hectare. The first class with the lower density of buildings ($\leq$2.5 buildings/ha) can be characterized as an isolated dwelling [32], and this covered the greatest part of the WUI zone, specifically 91.55% of the WUI category I area (Table 2). The second class (2.5–6 buildings/ha) can be characterized as scattered habitat, and covered 5.66% of the WUI area, and the third class, with high building density (6–50 buildings/ha), covered 2.79% of the WUI area.

**Table 2.** Distribution of **WUI zone (WUI category I)** based on the building density and the type of forest vegetation, and the percentages of the WUI area they cover.

| Vegetation Type | Building Density | | |
|---|---|---|---|
| | $\leq$**2.5 Buildings/ha** | **2.5–6 Buildings/ha** | $\geq$**6 Buildings/ha** |
| *Pinus brutia* forest | 22.15% | 0.26% | 0.05% |
| Evergreen vegetation | 26.92% | 0.64% | 0.22% |
| Burnt area/low vegetation | 12.94% | 0.25% | 0.03% |

In terms of the percentage of the WUI covered by a combination of buildings and different types of forest vegetation, it can be seen in the Table 2 that only a small percentage of the buffer zone has a building density over 2.5 buildings/ha, and it is characterized by the flammable vegetation type (Figure 4).

Concerning the distribution of the WUI area within the vulnerable zone of 100 m around the buildings, in terms of building density and the type of the forest vegetation and the percentages of the WUI area they cover, it can be seen in Table 3 that only a small part is characterized by the *Pinus brutia* forest. The other areas are characterized by evergreen vegetation and all classes of building density and, as such, have a lower risk of fire than the previous areas, and their economic values vary depending on the building density. Finally, there are areas close to buildings (less than 100 m), where there is low vegetation, and the building density is quite high.

**Table 3.** Distribution of the **WUI area** within the zone **of 100 m** around the buildings, based on the building density and the type of the forest vegetation, and the percentages of the WUI area they cover.

| Vegetation Type | Building Density | | |
|---|---|---|---|
| | ≤2.5 Buildings/ha | 2.5–6 Buildings/ha | ≥6 Buildings/ha |
| *Pinus brutia* forest within the distance of 100 m around the houses and infrastructure | 6.37% | 1.97% | 0.81% |
| Evergreen vegetation within the distance of 100 m around the houses and infrastructure | 42.67% | 23.62% | 15.17% |
| Burnt area/low vegetation | 46.68% | 26.53% | 17.17% |

*3.2. Ecological Characteristics of the WUI Area of the City of Thessaloniki*

According to the analysis of land cover within the WUI area, the forest vegetation of Thessaloniki can be distinguished into the following three main forest classes (Table 4): (a) high *Pinus brutia* even-aged forest coming from old reforestation, (b) evergreen vegetation of the Ostryo-Carpinion zone (pseudomaquis), and (c) the burned part of the forest with the vegetation that was established (naturally or artificially) after the 1997 fire. Based on the analysis of the collected field data, the *Pinus brutia* forest was characterized by low tree density (320 stems per hectare), with an average ground cover 50–70%. The average height of the trees was 11.5 (±0.6) m, and the average diameter at breast height was 28.3 (±1.4) cm (data not presented in a Table). The shrub storey was locally abundant, with the species *Quercus coccifera* and *Phillyrea latifolia* dominating, with percentages of 12.1% and 4.0%, respectively. In the burned forest, the predominant species was also *Pinus brutia*, but with a significantly lower cover percentage of 20.3% and a high participation of the evergreen species *Quercus coccifera* and *Phillyrea latifolia*. Pubescent oak (*Quercus pubescens*) and other planted species (e.g., *Cupressus sempervirens*, *Fraxinus ornus*, etc.) had a significant presence in the places where reforestation took place (Table 4). The mean height of the dominant species reached 4.5 m for *Pinus brutia*, while in the other species it was much lower (2.3 m) (data not presented in a table). In the areas where the evergreen vegetation prevailed, the dominant species was the kermes oak (*Quercus coccifera*), with a significantly higher percentage of 38.3%, while a high percentage of the species *Phillyrea latifolia* and *Cistus incanus* was also recorded (6.5 and 7.1%, respectively). The height of the dominant species (*Quercus coccifera* and *Phillyrea latifolia*) ranged between 3 and 4 m, while their DBH was less than 4 cm.

According to the analysis of the soil samples taken, the soil characteristics in the area of WUI around Thessaloniki showed generally small differences in terms of physical and chemical properties. However, some were found, with significant differences between the vegetation types (Table 5), especially between *Pinus brutia* forest (both burned and non-burned) and evergreen vegetation. These were related to soil depth and organic matter and nitrogen of the surface layer, which were found to be significantly lower in evergreen vegetation. A shallow soil depth was observed in all cases (a soil depth of less than 30 cm), especially in the southern and western exposures. Soils in all cases were neutral (pH ranged from 7.0 to 7.4), with moderate concentrations of organic matter and nitrogen in the surface layers.

**Table 4.** Canopy cover (%) of the dominant woody species in the three types of vegetation. Values represent means and standard errors of mean. Values in the same row followed by different letters indicate significant differences between them.

| Species | Plant Canopy Cover (%) | | |
|---|---|---|---|
| | *Pinus brutia* Forest | Burnt Forest Area | Evergreen Vegetation |
| *Pinus brutia* | 54.3 (±7.0) a | 20.3 (±2.6) b | 0.0 (±0.0) c |
| *Quercus coccifera* | 12.1 (±2.2) b | 15.1 (±4.0) b | 38.3 (±5.2) a |
| *Phillyrea latifolia* | 4.0 (±1.2) | 4.5 (±1.3) | 6.5 (±1.1) |
| *Cistus incanus* | 1.6 (±1.0) c | 11.3 (±2.8) a | 7.1 (±2.7) b |
| *Anthyllis hermaniae* | 1.1 (±0.8) | 3.1 (±1.2) | 2.1 (±1.0) |
| *Asparagus acutifolius* | 1.2 (±0.4) | 1.8 (±0.7) | 1.1 (±0.4) |
| *Sarcopoterium spinosum* | 1.3 (±0.9) | 2.6 (±0.6) | 1.2 (±0.6) |
| *Crataegus monogyna* | 0.7 (±0.2) | 0.8 (±0.2) | 1.2 (±0.3) |
| Other species | 3.5 (±1.4) | 2.8 (±1.2) | 3.7 (±1.6) |
| Total | 79.8 (±8.5) a | 62.3 (±7.7) b | 61.2 (±8.3) b |

**Table 5.** Soil characteristics in the three main vegetation classes of the studied WUI area. Values in the same row followed by different letters indicate significant differences between them.

| Vegetation Class/Soil Characteristics | Soil Layer | *Pinus brutia* Forest | Evergreen Vegetation | Burned Forest with the Vegetation Established after Fire in 1997 |
|---|---|---|---|---|
| Soil depth (cm) | | 27.5 a | 18.7 b | 27.3 a |
| Organic matter (%) | 0–15 cm | 3.4 ab | 3.1 b | 3.8 a |
| | 15–30 cm | 1.4 | 1.2 | 1.5 |
| pH | 0–15 cm | 7.3 | 7.2 | 7.2 |
| | 15–30 cm | 7.1 | 7.1 | 7.0 |
| Nitrogen (%) | 0–15 cm | 0.21 a | 0.18 b | 0.22 a |
| | 15–30 cm | 0.10 | 0.09 | 0.13 |
| Soil texture | 0–15 cm | SL * | SL | SL |
| | 15–30 cm | | | |

* Here, SL = sandy loam.

### 3.3. Human Pressure in the WUI Area of the Thessaloniki Metropolitan Area

The distinguished WUI area belong to four municipalities, as shown in Table 6. More than half of the WUI area belongs to the municipality of Pylaia-Hortiatis, indicating the high responsibility that this municipality should have in terms of fire management. On the other hand, the municipality of Thessaloniki is the most populated, which means that the behavior of its citizens may be the most important factor in lowering future fire risk. It must be noticed that population trends for the last 10 years greatly differ between the municipalities; two of them presented an increase, while the other two showed a decrease. The highest increase (+44.44%) was observed in the municipality of Pylaia-Hortiatis, a peripheral municipality, while the central municipality of Thessaloniki showed the largest decrease (−15.63%), indicating the different dynamics of each municipality, probably according to their position in the metropolitical area of Thessaloniki. According to the questionnaire, there were some differences observed between the citizens' rate of visiting the peri-urban forest areas of the three adjacent municipalities that, in combination with the respective population of the municipalities and the WUI area corresponding to them, result in a different level of human pressure in the WUI areas of each municipality (Table 6).

The most affected WUI area is that belonging to the municipality of Thessaloniki, with over 15 times more visits than the WUI areas belonging to other municipalities, while the WUI area of the municipality of Pylaia-Hortiatis was subjected to the lowest human population pressure. However, based on the population trends, this pressure is anticipated to greatly increase in the case of the municipality of Pylaia-Hortiatis.

**Table 6.** Human population pressure (in terms of number of human visits per year per hectare) in the **WUI** area of Thessaloniki. Note that we consider visiting turnovers similar between the municipalities. Values in the same column followed by different letters indicate significant differences between them.

| Municipality | Total Population | Population Trends for the Last 10 years | Adjacent WUI Area ha | Average Number of a Citizen Visiting WUI $y^{-1}$ | Total Number of Visiting WUI per Year | Total Pressure N $y^{-1}$ $ha^{-1}$ |
|---|---|---|---|---|---|---|
| Thessaloniki | 325,182 | −15.63% | 84.58 | 12.8 a | 4,162,330 a | 49,211.8 a |
| Neapolis-Sykeon | 84,741 | −5.08% | 487.91 | 10.1 b | 855,884 b | 1754.2 c |
| Pylaia-Hortiatis | 70,653 | +40.44% | 1312.64 | 9.6 b | 678,269 b | 516.7 d |
| Pavlou Mela | 99,245 | +13.31% | 318.86 | 9.9 b | 982,526 b | 3081.4 b |

Another important point we should consider is the distribution of the WUI categories in the administration borders of the four adjacent municipalities. According to Figure 4, the greatest part of the most populated WUI area, for both WUI categories, with high building densities, belongs to the municipality of Pylaia-Hortiatis. These areas also include high sensitivity infrastructure, such as healthcare (e.g., private and public hospitals, and clinics for children with special needs), private schools, etc.

## 4. Discussion

During the last decades, areas where urban space and wildland meet or intermingle have greatly increased worldwide, and these present a high risk of forest fires [36]. Thus, wildfire prevention measures are needed to reduce the vulnerability of these areas [5]. Understanding the dynamics of these wildland–urban interfaces (WUIs) in Mediterranean landscapes is particularly challenging because of multiple biophysical factors (dry or arid climate, low-quality soils, vegetation cover, and anthropogenic pressure) [37]. The combination of multiple anthropogenic causes of ignition and the mixing of flammable forest vegetation with human infrastructure results in the creation of a dangerous fire environment in which human lives and property are directly threatened.

This paper determines, maps, and characterizes in detail the WUI areas at a local spatial scale, considering the spatial arrangement of buildings and ecological forest characteristics, and suggests fire-prevention measures for these interface areas. The analysis of the results showed that the forest–settlement mixing zone (WUI), which is in contact with the metropolitan city Thessaloniki, occupies an area of 2203.98 hectares and that it is bounded between the peri-urban forest and the city. The WUI area appears at a low altitude ranging between 35 and 370 m, in medium inclinations, and at a short distance from the sea. The fact that there are no large slopes is a positive, since a slope is one of the most significant influencing factors affecting the spread of wildfires [9,38]. In the WUI study area, the building density varies greatly (0–50 buildings per ha), which is also supported by other studies [39,40], and was, thus, categorized into the following three classes [16,32]: the class of low density of buildings, which covers a percentage of 91.55%, the medium density class, which covers 5.66% of the terrain, and the high class, which covers 2.79% of the total area of the zone. According to Chas-Amil et al. [39] WUI areas with densely clustered buildings (4.7 buildings per ha) surrounded by forestlands have the highest fire ignition density.

The main vegetation that appears within the WUI area belongs to the following three forest classes: the pure mature even-aged *Pinus brutia* forest, the evergreen vegetation of the Ostryo-Carpinion zone (pseudomaquis), and the burned part of the forest with the vegetation that was installed after the 1997 fire event that burnt half of the forest and caused

negative impacts on ecosystem functions and local biodiversity [41,42]. All three of the above vegetation types belong to the Mediterranean forest ecosystems where wildfires are a common phenomenon [2,43], especially when these forest ecosystems are close to urban areas or intermingle with them.

The conflict between human activities and the environment already observed in these areas is expected to become even more intense in the future with the expansion of residential development in areas of high natural aesthetic or biodiversity value, as many of these are forest areas [36,44–46]. Human population pressure is also high, especially in the WUI area belonging to the municipality of Thessaloniki, compared to the other three adjacent municipalities, Neapolis-Sykeon, Pylaia-Hortiatis, and Pavlou Mela. This results in an increase in the risk of wildfires. Thus, it is suggested to plan specific pre-fire prevention measures against a possible wildfire that can help to a implement a better response based on a strategically multiple treated forest stands across the landscape [47], using computer modeling to determine where on a landscape to apply hazard fuel reduction treatment [9,48]. A silvicultural fire prevention treatment was already applied in the studied area in 2017 in the mature *Pinus brutia* forest, aimed to counter the increase in crown base height and to reduce the amount of ladder fuels.

Based on the findings of the current study, a great part of WUI is dominated by young *P. brutia* stands, established after the fire which destroyed half of the forest, characterized by a dense overstorey of young even-aged stands with an average height of 4.5 m, and high participation of the evergreen broadleaved species *Quercus coccifera* and *Phillyrea latifolia* in the shrub story. Additionally, the planted tree species *Quercus pubescens*, *Cupressus sempervirens*, *Fraxinus ornus*, etc. have a significant presence in the places where reforestation took place. The other part of the WUI area is dominated by evergreen vegetation, mainly by the evergreen oak *Quercus coccifera*, with an average height of 3–4 m, accompanied by the species *Phillyrea latifolia* and *Cistus incanus*. All these types of forest ecosystems are very flammable [41,43]; they are characterized by a continuous aerial fuel in all the areas they occur and, thus, special silvicultural treatments must be applied, resulting in fuel reduction and minimizing fire danger. However, an appropriate design of spatial arrangement of the treatments should be made, taking into consideration vegetation distribution, as well as building presence, in a way that will effectively prevent catastrophic fires or at least reduce damages from a possible fire event [9].

In order to improve safety for residents and increase the resilience of forests subjected to fire within the whole wildland–urban interface area of the Thessaloniki metropolitan area, a planning of a special social framework should be designed, defining protocols for responsible authorities and strict specific rules for citizen behavior. Additionally, some fuel-reduction silvicultural treatments are strongly recommended [9,49–53], as follows:

- Maintaining low levels of ladder fuel loading should be a primary management concern;
- Reduction in crown bulk density and an increase in crown base height through pruning and low thinning in the unburned pure mature forest stands of *P. brutia* should be carried out;
- Vegetation clearing, pruning, and thinning should take place in the young *P. brutia* forest, as well as in area occupied by the evergreen vegetation;
- The recommended target stand structure for WUI is mixed stands with components of broadleaves and, thus, complementary plantings with broadleaves are suggested.

## 5. Conclusions—Special Management Implications Measures for WUI Areas

### 5.1. Areas' Prioritization for Taken Intensive Fire Protection Measures

Prioritization should be given to the following categories:

WUI I: The areas with *Pinus brutia* forest and/or high building density (second and third density class) should be a priority in the design of fire protection actions, as they, on the one hand, pose a significant risk of fire due to the high presence of forest vegetation and, on the other hand, the area's value is very high because of the high building density.

WUI II: The vulnerable zone of 100 m around the buildings, and especially the small part characterized by *Pinus brutia* forest and/or high building density (second and third density class), should be a priority in the design of fire protection actions, as these areas, on the one hand, pose a significant risk of fire due to the high presence of forest vegetation and, on the other hand, are very high value areas because of the high building density.

Additionally, the areas close to buildings (less than 100 m), where there is a low vegetation, and the building density is quite high should be considered. These areas should be a priority in the design of fire protection, as there is a significant risk of fire and damage to homes, as forest vegetation occurs at a very close distance from homes and other infrastructure.

*5.2. Determination of the Specific Fire Protection Measures*

In the above priority areas, a set of special silvicultural fuel treatments must be applied to aim for fuel reduction and minimized fire danger. These include the following:

- In areas with mature *Pinus brutia* trees, the reduction in crown bulk density and increase in crown base height through pruning and low thinning should be performed [48–51]. Pruning up to at least to 1/3 of total tree height should be carried out. Thinning should be performed so that the crowns of different trees do not intermix with each other.
- In the young *P. brutia* forest, thinning, tree pruning, and vegetation clearing should be undertaken.
- In areas occupied by evergreen vegetation, vegetation clearing, pruning, and thinning should be carried out.

Except for silvicultural fuel treatments, some improvements in fire-suppression infrastructures would help in fire risk reduction. These include improvement of the existing forest road network to provide accessibility and access to firefighting vehicles, etc., mainly in high fire risk areas. Placing information signs on high-traffic roads passing through high-risk areas, as well as at intersections of the main forest road network, and the construction of fire outposts, will also assist in fighting fires. Additionally, information and the involvement of the local community could contribute to risk reduction, such as via meetings and seminars for information and familiarization with the staff of the Fire Service. It is also important to employ young firefighters with knowledge of the special conditions of the area (vegetation, habitats, value of the area, high risk zones and WUIs, topography, relief, the road network and its accessibility, the network of water intake points and its adequacy, etc.).

Finally, the results from our study could help managers in decision-making for specific fuel treatment and thinning prescription development. However, extending the monitoring of settlement dynamics and human population pressure, as well as vegetation changes in several WUI areas with different ecological conditions, under different climate crisis scenarios, is required in order to support effective sustainable management of Mediterranean suburban forests.

**Supplementary Materials:** The following supporting information can be downloaded at: https://www.mdpi.com/article/10.3390/fire5050159/s1, Table S1: Confusion matrix of the accuracy assessment of the area.

**Author Contributions:** Conceptualization, P.G.; methodology, P.G., N.O., and M.T.; software, N.O.; validation, P.G., and M.T.; formal analysis, M.T.; investigation, P.G., M.T., and N.O.; resources, P.G. and N.O.; data curation, M.T. and N.O.; writing—original draft preparation, P.G.; writing—review and editing, M.T. and N.O.; visualization, N.O.; supervision, P.G.; project administration, PG. All authors have read and agreed to the published version of the manuscript.

**Funding:** There was no funding for this research.

**Institutional Review Board Statement:** Not applicable.

**Informed Consent Statement:** Not applicable.

**Conflicts of Interest:** The authors declare no conflict of interest.

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
