# Peer review of "Small-Scale Analysis of Characteristics of the Wildland–Urban Interface Area of Thessaloniki, Northern Greece"

_fire, doi:10.3390/fire5050159_

Round 1
Reviewer 1 Report (Previous Reviewer 1)
The authors have done an excellent job in tackling the reviewer's queries. I find the manuscript greatly improved and ready to be accepted for publication—no more questions from me.
Author Response
Many thanks to your positive comments.
Reviewer 2 Report (Previous Reviewer 2)
No further comments.
Author Response
Thank you for your positive comments.
Reviewer 3 Report (Previous Reviewer 3)
This work is a thorough study of the WUI around Thessaloniki. Besides its academic interest, the work can be potentially useful for policy makers, the fire service and the inhabitants. Please find some final language comments.
L24: and low thinning, aiming at tree crowns of mature Pinus brutia trees not being in contact with one another.
L24 – 26: In the young P. brutia forest, as well as the evergreen vegetation, thinning, tree pruning, and vegetation clearing are recommended. (What are the differences in the treatment recommendations between P. brutia forest and the evergreen vegetation? It looks for me that the same recommendations (thinning, pruning, vegetation clearing) are given, only in a different order.)
L36: been given much attention during the last years, mainly …
L52: However, WUI areas are among the areas most vulnerable to wildfires in many countries …
L78 – 81: Though clear methodological approaches for mapping WUI exist [Ref?], the methods do not provide the ecological precision required for effective management interventions.
L87: Unfulfilled sentence. Or does it only lack the final dot?
L103: replace “consist” with “entail”, and place DOT after [13]. Half of Thessaloniki´s peri-urban forest was burnt in 1997, causing great ecological damage to the area and the city.
L106: replace “depressed” with “suppressed”.
L135 – 137: re-write. (The result … study.)
L168 – 170: re-write. (Also for … areas (Figures 1, 3)).
L265 - 266: Particularly, low slope values were present (0 – 15%) in 88.68% of the total WUI area, while slopes >30% existed in only 0.16% of the WUI area.
L 381: is particularly challenging
L407: especially when these forest ecosystems are close to urban areas or intermingle with them.
L471: Thinning so that the crowns of different trees do not intermix with each other.
Author Response
Authors would like to express their thanks to the Reviewer valuable comments.
All the comments were considered in the Revision of the manuscript (Text in red color).
This manuscript is a resubmission of an earlier submission. The following is a list of the peer review reports and author responses from that submission.
Round 1
Reviewer 1 Report
In this study, the authors tackle a very interesting and important aspect of wildfire science: understanding and identifying the wildland-urban interface in a region for applying better fire management to combat wildfire. They have identified the wildfire potential and mapped the area of Thessaloniki, northern Greece in different WUI zones by combining GIS data, examining the housing density, existing vegetation, and associated human population pressures in the region. They have also suggested some silvicultural treatments and other improvements that can be applied for increasing the resilience of the area to wildfire. Wildfires are increasing globally, especially in the area of Mediterranean regions that are vulnerable to wildfires due to climate change, land-use change, and more people living beside the forest. Like this study, identifying the proper WUI zones are necessary to focus on and apply fire management approaches that can reduce the damage from wildfires and protect human lives and properties.
The manuscript is well structured and written for the most part. I have only one concern regarding the data analysis. I am wondering why the authors did not perform any statistical test to compare ecological variables (canopy covers, vegetation types, soil properties) measured here as well as data on different aspects of human population pressures between different municipalities collected by questionnaires. They can easily perform an ANOVA test to properly describe those data in the results section, particularly how the variables vary between WUI zones, or how the canopy cover, building density, and soil characteristics vary statistically.
I have also made plenty of minor corrections and some comments in the pdf file (attached here), please go through them.

Author Response
Reviewer overall comments
In this study, the authors tackle a very interesting and important aspect of wildfire science: understanding and identifying the wildland-urban interface in a region for applying better fire management to combat wildfire. They have identified the wildfire potential and mapped the area of Thessaloniki, northern Greece in different WUI zones by combining GIS data, examining the housing density, existing vegetation, and associated human population pressures in the region. They have also suggested some silvicultural treatments and other improvements that can be applied for increasing the resilience of the area to wildfire. Wildfires are increasing globally, especially in the area of Mediterranean regions that are vulnerable to wildfires due to climate change, land-use change, and more people living beside the forest. Like this study, identifying the proper WUI zones are necessary to focus on and apply fire management approaches that can reduce the damage from wildfires and protect human lives and properties.
Authors’ general response
Thank you very much for your valuable comments that greatly helped us in manuscript improving.
All your comments were considered in manuscript revision.
All the text modification is indicated in green color in the revised manuscript.
Reviewer comment
The manuscript is well structured and written for the most part. I have only one concern regarding the data analysis. I am wondering why the authors did not perform any statistical test to compare ecological variables (canopy covers, vegetation types, soil properties) measured here as well as data on different aspects of human population pressures between different municipalities collected by questionnaires. They can easily perform an ANOVA test to properly describe those data in the results section, particularly how the variables vary between WUI zones, or how the canopy cover, building density, and soil characteristics vary statistically.
Authors’ responses
The requested ANOVA was performed; data are shown in the relevant tables (Tables 4,5 and 6) in the revised manuscript, and in the relevant text (e.g. lines 294, 301-302, 311-314).
Reviewer comment
I have also made plenty of minor corrections and some comments in the pdf file (attached here), please go through them.
Authors’ responses
Thank you very much for the apt corrections and the constructive comments. All the suggestions were applied (text with green color in the revised manuscript).
A new figure (Figure 4) was added, according to the suggestions.
Reviewer 2 Report
This paper used the OBIA mapped results to analyze the human and ecological characteristics of the Wildland Urban Interface area in Thessaloniki, northern Greece. The qualitative analysis was conducted on human pressure and ecological characteristics, in order to inform suggestions for fire prevention. The article lacks innovation. The authors just implemented the OBIA algorithm to classify the city of Thessaloniki, then combined the results of the classification to quantitatively describe two features of WUI, and tried to make recommendations for fire protection through these descriptions. Article Lacks Quantitative Analysis of Fire Hazard that may result from urban expansion caused Wildland Urban Interface area. If the novelty of this paper is just a case study of the city of Thessaloniki without any generalized quantitative findings, I would not recommend it for publication.
Author Response
Reviewer overall comments
This paper used the OBIA mapped results to analyze the human and ecological characteristics of the Wildland Urban Interface area in Thessaloniki, northern Greece. The qualitative analysis was conducted on human pressure and ecological characteristics, in order to inform suggestions for fire prevention. The article lacks innovation. The authors just implemented the OBIA algorithm to classify the city of Thessaloniki, then combined the results of the classification to quantitatively describe two features of WUI, and tried to make recommendations for fire protection through these descriptions. Article Lacks Quantitative Analysis of Fire Hazard that may result from urban expansion caused Wildland Urban Interface area. If the novelty of this paper is just a case study of the city of Thessaloniki without any generalized quantitative findings, I would not recommend it for publication.
Authors’ responses
Thank you very much for your valuable comments that greatly helped us in manuscript improving.
All your comments were considered in manuscript revision.
Thus, we made an effort to revise the paper in order to match with your suggestions. Analytically, we modified the text, in many points along the manuscript, from the Introduction to Discussion; we added a new figure (Figure 4) showing the results of our analysis, and performed a further analysis of human pressure and population dynamics, in order to produce a quite integrated approach for WUI areas. In fact, our approach was based on the analytical study of all the structural field characteristics of a WUI area, including ecological, building, human pressure, and their spatial differentiation. We focused on real data collection and analysis than a modeling approach.
All the text modification is indicated in green color in the revised manuscript.
Reviewer 3 Report
Review report, “Small-scale analysis of characteristics of the Wildland-Urban Interface area of Thessaloniki, northern Greece”.
Overall comments
Mapping and analysing WUIs requires detailed classification of quantity and type of vegetation, as well as type and distribution of structures over large areas. Maps at local scale are therefore needed. The authors have mapped and analysed the WUI-area in Thessaloniki, using Sentinel 2 satellite images, Very Large Scale Orthophotos (VLSO) and Object Based Image Analysis (OBIA), combined with field studies, to categorize landcover. The intention was to identify valuable areas exposed to high wildfire hazard. The results may inform policy makers and citizens and guide them to prioritise fire preventive interventions.
The use of satellite images combined with GIS systems and field data analysis for land cover characterization is well established. The awareness of fire hazard in the WUI has dramatically increased in recent years worldwide, largely because of the high toll in human lives, mainly in Australia, California, and Euro-Mediterranean countries. Though the societal impact of research into these matters is high, the novelty of the research may appear as average (and after some years it may perhaps also be classified as “low”), as the techniques become standardized.
The authors used redundant sources i.e., maps of forest and grasslands by the Hellenic Cadastral Organization, Corine 2018 for Greece, maps from the Hellenic Forest Service and CLMS to avoid misclassifications. The methodological part of the work is strong. Additionally, the ecological characteristics of the WUI were investigated with 10 sample plots in each of the three prevailing nature types; pine forest, shrubland, area burned in 1997. Information about vegetation species abundance (Table 4) and soil characteristics (Table 5) is presented. It was also investigated how often residents in the different municipalities neighbouring to the WUI visit the forest (Table 6). Combining information on building density with nature type may improve understanding of WUI-fire potential.
The article suggests some silvicultural treatments in the Conclusion session, without connecting them to the characterization of the WUI done earlier. All suggested measures are reasonable, however general. I suggest that the authors move those possible measures to chapter 4, Discussion, and thereafter suggest specific measures for the most exposed areas as identified in Table 2. Otherwise, there is little connection between the analysis of the WUI (title) and the suggested measures (conclusions). It could be of great help to the forest service with specific recommendations of which treatment to apply in which of the three main vegetation types, and directly point to the areas that represent a WUI fire hazard.
The article is well-written. However, a language-check by professional (or native English-speaking person) will remove the many small errors throughout the manuscript. These are not commented in detail since there are too many of them.
Detailed comments
Lines 20 and 21: “A set of fire prevention silvicultural treatments are suggested for mitigating the fire danger in the area, accompanied with appropriate human awareness actions and involvement of the local society.” I can not see any specific treatments being suggested.
Line 89: the ultimate goal was the utilization (of the results from the present research) in designing specific fire-protection measures. Which ones fit where??
Line 146: WUI category I is defined in the Figure text, Figure 1 (a 500 m wide zone). WUI category II is defined in line 173 and in Figure 3 as a nuanced concept with different building densities. Please clarify or check for eventual inconsistencies.
Line 177: density 0 – 0.0625 (up to 1 building/16 ha). In line 179 it is commented: “However, the first class was recorded only in the WUI category I”. Were isolated houses present in the 500 m zone, while it becomes denser in other areas of WUI category II? Please check.
Line 222: “as the confusion matrix shows (Table S1). Where do we find this table?
Lines 251 and 252 present an important finding, of areas with valuable buildings being in fire hazardous environment, while the next lines suggest actions to reduce the risk. Perhaps a new figure could be introduced to pinpoint where these areas are, instead of only saying that they only represent a small percentage of the buffer zone (see Table 2)?
Lines 264 – 267 can be deleted, as they repeat the text from lines 253 – 256.
Line 335, “and suggests fire-prevention measures for these interface areas.” Which measures? If you do not suggest any specific measures, which you deem are especially suitable for exactly this area, you may rewrite the sentence like: “and suggests that local stakeholders identify and implement fire-prevention measures for these interface areas.”
Lines 365 – 367. The information given, about applied treatment in 2017, reducing ladder fuel and increasing the crown – base height in the mature Pinus brutia forest is uplifting. Perhaps additional focus must be given to systematic / regular treatment, to avoid new encroachment.
Line 368 – 382. These lines focus on the area burned and 1997 and observe that though several species are present, the vegetation has dense overstorey with continuous aerial fuel, and suggest silvicultural treatments for fuel reduction (for example thinning, firebreaks, etc.)
Author Response
Reviewer overall comments
Mapping and analysing WUIs requires detailed classification of quantity and type of vegetation, as well as type and distribution of structures over large areas. Maps at local scale are therefore needed. The authors have mapped and analysed the WUI-area in Thessaloniki, using Sentinel 2 satellite images, Very Large Scale Orthophotos (VLSO) and Object Based Image Analysis (OBIA), combined with field studies, to categorize landcover. The intention was to identify valuable areas exposed to high wildfire hazard. The results may inform policy makers and citizens and guide them to prioritise fire preventive interventions.
The use of satellite images combined with GIS systems and field data analysis for land cover characterization is well established. The awareness of fire hazard in the WUI has dramatically increased in recent years worldwide, largely because of the high toll in human lives, mainly in Australia, California, and Euro-Mediterranean countries. Though the societal impact of research into these matters is high, the novelty of the research may appear as average (and after some years it may perhaps also be classified as “low”), as the techniques become standardized.
The authors used redundant sources i.e., maps of forest and grasslands by the Hellenic Cadastral Organization, Corine 2018 for Greece, maps from the Hellenic Forest Service and CLMS to avoid misclassifications. The methodological part of the work is strong. Additionally, the ecological characteristics of the WUI were investigated with 10 sample plots in each of the three prevailing nature types; pine forest, shrubland, area burned in 1997. Information about vegetation species abundance (Table 4) and soil characteristics (Table 5) is presented. It was also investigated how often residents in the different municipalities neighbouring to the WUI visit the forest (Table 6). Combining information on building density with nature type may improve understanding of WUI-fire potential.
Authors’ responses
Thank you very much for your valuable comments that greatly helped us in manuscript improving.
All your comments were considered in manuscript revision.
All the text modification is indicated in green color in the revised manuscript.
Reviewer comment
The article suggests some silvicultural treatments in the Conclusion session, without connecting them to the characterization of the WUI done earlier. All suggested measures are reasonable, however general. I suggest that the authors move those possible measures to chapter 4, Discussion, and thereafter suggest specific measures for the most exposed areas as identified in Table 2. Otherwise, there is little connection between the analysis of the WUI (title) and the suggested measures (conclusions). It could be of great help to the forest service with specific recommendations of which treatment to apply in which of the three main vegetation types, and directly point to the areas that represent a WUI fire hazard.
Authors’ responses
Thank you very much for your valuable comments. Accordingly, we moved “those possible measures” to chapter 4, Discussion as it is suggested, and thereafter we recommended specific measures for the most exposed areas as identified in Table 2 in the next section 5. Conclusions – Special management implications measures for WUI areas (subsections, 5.1. Areas’ prioritization for taken intensive fire protection and “5.2. Determination of the specific fire protection measures”).
Reviewer comment
The article is well-written. However, a language-check by professional (or native English-speaking person) will remove the many small errors throughout the manuscript. These are not commented in detail since there are too many of them.
Authors’ responses
Thank you very much for your valuable comments. A language-check by professional was performed. All the text modification is indicated in green color in the revised manuscript.
Detailed comments
Reviewer comment
Lines 20 and 21: “A set of fire prevention silvicultural treatments are suggested for mitigating the fire danger in the area, accompanied with appropriate human awareness actions and involvement of the local society.” I can not see any specific treatments being suggested.
Authors’ responses
Thank you for the suggestion. We added some (the main) specific treatments (lines 22-26), in the revised manuscript.
Reviewer comment
Line 89: the ultimate goal was the utilization (of the results from the present research) in designing specific fire-protection measures. Which ones fit where??
Authors’ responses
We greatly modified the text in order to cover the gap; please see the relevant subsections, 5.1. Areas’ prioritization for taken intensive fire protection, and “5.2. Determination of the specific fire protection measures”, in the revised manuscript.
Reviewer comment
Line 146: WUI category I is defined in the Figure text, Figure 1 (a 500 m wide zone). WUI category II is defined in line 173 and in Figure 3 as a nuanced concept with different building densities. Please clarify or check for eventual inconsistencies.
Authors’ responses
We carefully checked for any inconsistencies.
Reviewer comment
Line 177: density 0 – 0.0625 (up to 1 building/16 ha). In line 179 it is commented: “However, the first class was recorded only in the WUI category I”. Were isolated houses present in the 500 m zone, while it becomes denser in other areas of WUI category II? Please check.
Authors’ responses
We carefully checked the suggestion.
Reviewer comment
Line 222: “as the confusion matrix shows (Table S1). Where do we find this table?
Authors’ responses
The table was submitted as Supplementary Table S1.
Reviewer comment
Lines 251 and 252 present an important finding, of areas with valuable buildings being in fire hazardous environment, while the next lines suggest actions to reduce the risk. Perhaps a new figure could be introduced to pinpoint where these areas are, instead of only saying that they only represent a small percentage of the buffer zone (see Table 2)?
Authors’ responses
Thank you very much for this specific suggestion. It was really helpful to us.
We added a new figure (Figure 4) showing all the relevant information.
Reviewer comment
Lines 264 – 267 can be deleted, as they repeat the text from lines 253 – 256.
Authors’ responses
We did it.
Reviewer comment
Line 335, “and suggests fire-prevention measures for these interface areas.” Which measures? If you do not suggest any specific measures, which you deem are especially suitable for exactly this area, you may rewrite the sentence like: “and suggests that local stakeholders identify and implement fire-prevention measures for these interface areas.”
Authors’ responses
According to this suggestion, and in combination with the previous about “the movement of the possible measures to chapter 4, Discussion, and thereafter recommend specific measures for the most exposed areas as identified in Table 2”, we made great changes in these two sections (4 and 5).
Additionally, we added two subsections in the Discussion (5.1. Areas’ prioritization for taken intensive fire protection and “5.2. Determination of the specific fire protection measures”).
All these changes are shown with green color in the revised manuscript.
Reviewer comment
Lines 365 – 367. The information given, about applied treatment in 2017, reducing ladder fuel and increasing the crown – base height in the mature Pinus brutia forest is uplifting. Perhaps additional focus must be given to systematic / regular treatment, to avoid new encroachment.
Authors’ responses
According to this suggestion, and in combination with the previous, we added two subsections in the Discussion (5.1. Areas’ prioritization for taken intensive fire protection and “5.2. Determination of the specific fire protection measures”).
Reviewer comment
Line 368 – 382. These lines focus on the area burned and 1997 and observe that though several species are present, the vegetation has dense overstorey with continuous aerial fuel, and suggest silvicultural treatments for fuel reduction (for example thinning, firebreaks, etc.)
Authors’ responses
In combination with the previous suggestions, we made great changes in the two sections (4 and 5), and we added the two subsections in the Discussion (5.1. Areas’ prioritization for taken intensive fire protection and “5.2. Determination of the specific fire protection measures”), in order to clarify the suggested silvicultural treatments per vegetation type.